# IHR-PVS National Bridging Workshops, a tool to operationalize the collaboration between human and animal health while advancing sector-specific goals in countries

**Guillaume Belot**[1]*, **François Caya**[2], **Kaylee Myhre Errecaborde**[1], **Tieble Traore**[3], **Brice Lafia**[4], **Artem Skrypnyk**[5], **Djhane Montabord**[6], **Maud Carron**[2], **Susan Corning**[2], **Rajesh Sreedharan**[1], **Nicolas Isla**[5], **Tanja Schmidt**[5], **Gyanendra Gongal**[7], **Dalia Samhouri**[8], **Enrique Perez-Gutierrez**[9], **Ana Riviere-Cinnamond**[9], **Jun Xing**[1], **Stella Chungong**[1], **Stephane de la Rocque**[1]

1 World Health Organization, Geneva, Switzerland, 2 World Organisation for Animal Health (OIE), Paris, France, 3 World Health Organization Regional Office for Africa, Brazzaville, Congo, 4 World Organisation for Animal Health (OIE) Regional Representation for Africa, Bamako, Mali, 5 World Health Organization Regional Office for Europe, Copenhagen, Denmark, 6 World Organisation for Animal Health (OIE) Sub-Regional Representation for Central Asia, Nur-Sultan, Kazakhstan, 7 World Health Organization Regional Office for South-East Asia, New Delhi, India, 8 World Health Organization Regional Office for the Eastern Mediterranean, Cairo, Egypt, 9 Pan American Health Organization / World Health Organization Regional Office for the Americas, Washington, United States of America

* belotg@who.int, guillaume.belot.pro@gmail.com

**Data Availability Statement:** All relevant data are within the paper and its Supporting information files.

## Abstract

Collaborative, One Health approaches support governments to effectively prevent, detect and respond to emerging health challenges, such as zoonotic diseases, that arise at the human-animal-environmental interfaces. To overcome these challenges, operational and outcome-oriented tools that enable animal health and human health services to work specifically on their collaboration are required. While international capacity and assessment frameworks such as the IHR-MEF (International Health Regulations—Monitoring and Evaluation Framework) and the OIE PVS (Performance of Veterinary Services) Pathway exist, a tool and process that could assess and strengthen the interactions between human and animal health sectors was needed. Through a series of six phased pilots, the IHR-PVS National Bridging Workshop (NBW) method was developed and refined. The NBW process gathers human and animal health stakeholders and follows seven sessions, scheduled across three days. The outputs from each session build towards the next one, following a structured process that goes from gap identification to joint planning of corrective measures. The NBW process allows human and animal health sector representatives to jointly identify actions that support collaboration while advancing evaluation goals identified through the IHR-MEF and the OIE PVS Pathway. By integrating sector-specific and collaborative goals, the NBWs help countries in creating a realistic, concrete and practical joint road map for enhanced compliance to international standards as well as strengthened preparedness and response for health security at the human-animal interface.

**Funding:** The organization of IHR-PVS National Bridging Workshops in countries was supported by many funders including the United States Defense Threat Reduction Agency (US-DTRA), the Global Partnership Program (GPP), the EU Commission's Directorate-General for International Cooperation and Development (DG DEVCO), the Russian Federation and the World Bank, among others.

**Competing interests:** The authors have declared that no competing interests exist.

## Introduction

In recent decades, the world has seen increasing emergence of infectious zoonotic diseases, including Severe Acute Respiratory Syndrome (SARS) in 2003, novel strains of Highly Pathogenic Avian Influenza (HPAI) in 1997 and in 2003, H1N1 Influenza pandemic in 2009, Middle Eastern Respiratory Syndrome Coronavirus (MERs-CoV) in 2012, Ebola virus in West, Central and Eastern Africa in 2014, 2018, 2019, 2020 and, most recently, the emergence of Severe Acute Respiratory Syndrome Coronavirus 2 (SARS-CoV-2), known as COVID-19 [1–4]. Out of all infectious organisms known to be pathogenic to humans, over 60% are zoonotic in nature. This figure increases to 75% when considering emerging pathogens [1], with a large proportion originating from wildlife [2, 5]. A variety of ecological and demographic factors, such as encroachment of human activities in the natural habitat of wild animals, intensified systems of agriculture, and increased volumes of traffic and trade are precipitating both the emergence of such diseases and their subsequent spread [6–9].

With these observations, the One Health concept, loosely defined as "*the collaborative efforts of multiple disciplines working locally, nationally, and globally, to attain optimal health for people, animals, and our environment*" [10], has gained great momentum over the past two decades as it becomes clear that collaboration between the different sectors can help countries to better face current and upcoming health threats [11–14].

The benefits of One Health go beyond emerging infectious diseases. It is also a much needed approach for other major global health challenges such as antimicrobial resistance [15, 16], food safety [17–19], bioterrorism [20], disaster recovery and response [12], and climate change [21] among others.

However, after decades of siloed medicine evolution, implementing this approach can incur many obstacles. Uncertain cost-effectiveness, availability of human resources, limited laboratory capacity, and long-standing barriers of privacy and distrust are some of the factors hindering the operationalization of the concept at country-level [22, 23].

To overcome these challenges, operational and outcome-oriented tools that engage and enable animal health and human health services to focus specifically on their collaboration, are required [24].

## Bridging capacity assessment and improvement frameworks between the human health and animal health sectors

Human health and animal health sectors use distinct evaluation frameworks to assess their existing capacities and to further improve them. This includes namely the IHR Monitoring and Evaluation Framework (IHR-MEF) for primarily public health, and the Performance of Veterinary Service (PVS) Pathway for animal health.

WHO Member States adopted a legally binding instrument, the International Health Regulations (IHR 2005) [25], for the prevention and control of events that may constitute a public health emergency of international concern. Through these regulations, State Parties to the IHR (2005) are required to develop, strengthen and maintain minimum national core public health capacities to early detect, assess, notify and rapidly respond to public health threats. Various assessment and monitoring tools have been developed as part of the IHR-MEF, including the States Parties Annual Report (SPAR) and the Joint External Evaluation (JEE) Tool. The SPAR is a self-assessment conducted by countries who are obligated, under the IHR (2005), to assess their core public health capacities and annually report the results to the IHR secretariat [26]. The JEE, on the other hand is run on a voluntary basis by Member States, and under the leadership of WHO. The JEE begins with a self-assessment by the country, using the JEE tool [27] which covers 19 technical areas to be assessed on a scale of 1-to-5 levels of advancement. A

panel of nominated international experts then conduct a one week in-country visit to meet with national stakeholders for a peer-to-peer review of the country's national capacities and to provide joint recommendations for their improvement. Both the SPAR and JEE contribute to the IHR-MEF.

On the other hand, the PVS Pathway was launched in 2007 by the World Organisation for Animal Health (OIE). It supports the sustainable strengthening of national Veterinary Services (VS) for greater compliance with OIE animal health standards [28] by providing countries with independent evaluations of their VS and tailored capacity-building activities [29]. The PVS Evaluation is a key component of the PVS Pathway, sometimes seen as the 'diagnosis' phase, and which paves the way for other support options such as the PVS Gap analysis which involves strategic planning and budgeting of VS activities. It is generally conducted through a 2-to-3 week in-country mission (up to 6 weeks for large countries) during which OIE trained PVS experts meet with national stakeholders to conduct an in-depth qualitative assessment of the country's Veterinary Services' strengths and weaknesses [30]. The mission uses the robust OIE PVS Tool, in which 45 Critical Competencies are to be assessed on a scale of 1-to-5 levels of advancement [31].

While both the OIE PVS Pathway and the SPAR/JEE do contain and promote some elements of transdisciplinary and intersectoral collaboration, lending to the concept of One Health, the need for a specific tool to operationalize the concept and support countries in improving and implementing collaborative efforts at the interface between humans and animals remained.

The OIE and WHO first conducted an analysis of the differences and synergies between the two frameworks and their associated tools in 2013. This initially focused on reviewing the linkages between the PVS Pathway approach as a whole and the IHR, including the annual reporting tool. This was first summarized in the 'WHO-OIE operational framework for good governance at the human-animal interface: Bridging WHO and OIE tools for the assessment of national capacities' [32]. By capitalizing on the strength of these existing sector-specific institutional frameworks, the two organizations jointly developed methods to facilitate communication between the animal health and human health sectors. This resulted in workshops organized in countries, allowing national counterparts to better understand both the IHR and the PVS, allowing them to agree upon priority needs and jointly elaborate on their bridging efforts [33]. Through a series of consultations, this fostered the development of the IHR-PVS National Bridging Workshops (NBWs).

The NBWs offer national stakeholders a unique opportunity to first 'diagnose' their existing collaboration challenges and gaps that exist between sectors, and then jointly develop actionable steps to strengthen collaboration that supports both PVS and IHR. Unlike other collaborative evaluation tools, NBWs link One Health actions directly to international policies and frameworks, providing a global approach that leverages shared actions across many countries.

In this article we introduce NBWs as a novel diagnostic and planning tool by describing its development, detailing its method and material and by discussing the preliminary outputs obtained from NBWs conducted in 32 countries.

## Method

Ethics statement: no research was conducted on human subjects or other animal subjects for the purpose of this article therefore no ethics approval was required. Participants to the workshops were invited and came in full consent. Their consent was not documented in any written way. Participants were informed as to the nature of their participation (fact-sheet, concept note, agenda) prior to coming to the workshop. In the opening session of every event, the first

presentation gave an overview of the method and process of the workshop and stated that results would be compiled in a report and posted on the WHO website and may later be used for further research and publication. The information obtained was recorded by the investigator in such a manner that the identity of the respondents cannot be readily ascertained, directly or indirectly through identifiers linked to the subjects.

Driven by OIE's and WHO's interest in better understanding and supporting countries to improve their IHR and PVS performances, the objective for NBWs was to develop a process which would give stakeholders from the human and animal health sectors an opportunity to discuss and evaluate their current collaboration and jointly plan for its strengthening. The purpose is not to provide them with recommendations or solutions, but to create an enabling environment during which they can identify what works best for them and how they can realistically improve the collaboration with nationally grounded solutions that fit their system and context.

The NBW method was developed through an iterative process involving two phases, each consisting of three in-country pilots. In phase one, an outline of activities that supported assessment and action planning was established and piloted in Azerbaijan, Costa Rica and Thailand as a proof of concept. In phase two, evaluative feedback from phase one facilitated the modification and strengthening of activities during pilots in Pakistan, Indonesia and Uganda. The strengthened approach enables countries to elaborate a comprehensive and very detailed joint Roadmap as a key output. Throughout both phases, different sessions and tools were trialed and tested, the results of which, along with feedback collected from participants and partners, were used to conduct evaluations after each pilot, to adapt the method and material and improve the tool.

## Phase one: Developing the concept

**Azerbaijan (2013) (46 national experts, 1.5 days).** In this first pilot, the method included presentations from the two sectors, along with a working exercise to look at the results of the respective assessments and discuss their linkages. The working group exercise consisted of facilitated discussion around a dozen key questions. The meeting was conducted over 1.5 days and included over 46 national experts.

The meeting was challenged by the low level of knowledge of the participants on the IHR (2005), the PVS Pathway and the associated tools. This shortcoming limited the ability for both sectors to engage in the discussion of outputs reported for either IHR or PVS. In the post-workshop survey, participants suggested a longer workshop, with more time for discussion and expressed high appreciation for the working group exercise.

**Thailand (2014) (59 national experts, 2 days).** Following the experience from Azerbaijan, some key changes to the method were implemented: the workshop was extended to two full days, a session was added to give more in-depth explanations on the IHR, the PVS and their connections, and a working group exercise was added to identify opportunities for synergetic actions between the two sectors. It was in the preparation for this second pilot that technical experts from OIE and WHO considered the opportunity to visually illustrate the interface of human and animal health in a matrix that reflected both IHR and PVS. This was one of the most important evolutions in the NBW process, resulting in the development of the IHR-PVS matrix (Fig 1) which crosses the indicators of the IHR (in rows) and the Critical Competencies of the PVS Pathway (in columns). This allowed participants to easily visualize all the connections between the two sectors and the two frameworks.

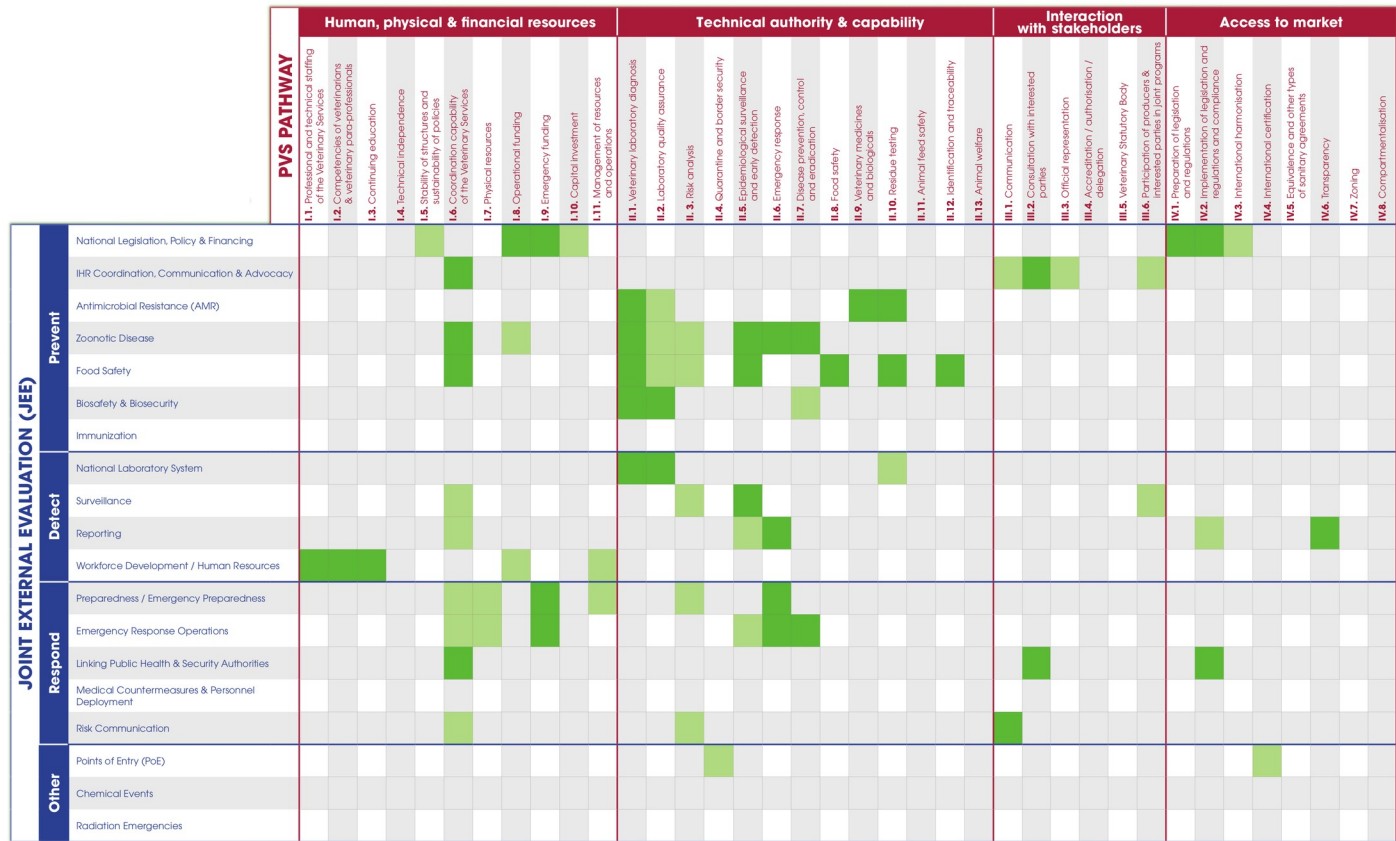

**Fig 1. The IHR-PVS matrix is a 5x3 meter presentation stand used by participants during the NBW.** The matrix crosses the indicators from the IHR-MEF in rows and the Critical Competencies of the PVS Pathway in columns. Two versions of the matrix exist regarding the IHR-MEF: one with SPAR indicators and one with JEE indicators. The matrix was produced in English, French, Russian and Spanish versions.

In the post-workshop survey, participants once again asked for a longer workshop with more time for discussion and group exercises. They also felt that there were too many presentations and found the working group sessions to be the most productive ones.

**Costa Rica (2016) (60 national experts, 2.5 days).** The third pilot incorporated a number of critical changes, including an increase to two-and-a-half days total duration, the reduction of the number of presentations, the replacement of some presentations by videos and the addition of a working group exercise using short outbreak scenarios (Table 1) to allow stakeholders

**Table 1. The five short disease scenarios used during the NBW pilot in Thailand.**

| Disease | Scenario |
|---|---|
| Rabies | A case of rabies, which has been confirmed in a dairy cow recently inseminated and regularly milked, generates panic in the population |
| H7N9 avian influenza | H7N9 was confirmed in a vet who returns from a conference in China and lives in the northern part of Thailand |
| Anthrax | Nine people showed identical anthrax-like lesions reported in a district hospital close to a border post. One is working in village slaughterhouse |
| Streptococcus suis | An exporting country suspects that a shipment of piglets to Thailand was contaminated with *Streptococcus suis* and entered into the market |
| Unknown disease | Private Veterinarian reports unusual mortality among piglets in a commercial farm. Workers on the farm also show illness |

to self-assess their level of collaboration for 15 key technical areas. This exercise resulted in the mapping of strong and weak areas in the collaboration, which participants used to draft the outline of a strategy to improve their inter-sectoral work.

When presented with simple scenarios, participants could more easily identify the strengths and weaknesses of their current collaboration and the conceptualization of joint activities was better facilitated.

This third pilot also highlighted the need for additional work sessions to be developed to transform the results of the discussions into an implementation plan.

## Phase two: Refining the tool

**Pakistan (2017), Indonesia (2017) and Uganda (2017).** After an in-depth look at the feedback collected from participants and partners during the three first pilots, a substantial revision of the material and method was conducted. Special focus was given to the development of three working exercises (i) to simplify the extraction of relevant information from the SPAR/JEE and PVS Pathway, (ii) to use the outcomes of the discussions to initiate a joint road-map, with a list of activities identified jointly by both sectors, and (iii) to fine-tune this road-map and discuss on the way forward while giving full ownership of the process and result to the country. The overall duration was increased to three days to optimally facilitate these changes.

The whole set of material, which included videos, activity cards and posters, was revised and adjusted. These updated method and material were tested in Pakistan (May 2017). This was the first time that a detailed roadmap was developed and then anchored in Kazakhstan's National Action Plan for Health Security (NAPHS).

Following next pilots in Indonesia (August 2017) and Uganda (September 2017), the method and material were further fine-tuned and ultimately finalized. Notable improvements included the development of participant handbook (S1 Appendix), the addition of a prioritization exercise (via an online vote when possible, or by using small stickers as votes) and an additional step where participants were invited to detail the operational process for the implementation of the joint activities they have identified.

At this stage, the method and material were considered complete and only very minor modifications were brought in the subsequent workshops, often just to adapt to different cultural contexts.

## Organization and facilitation of NBWs

The roll-out of NBWs is undertaken on a voluntary request from countries. Organization of the workshop begins when one or both of relevant Ministries makes an official request to either WHO or OIE. Once requested, NBWs are facilitated by at least two lead facilitators from both WHO and OIE. Country demands for NBWs exceeded expectations and the number of trained facilitators in the core team quickly became insufficient. Regional facilitators were therefore trained in both organizations for the roll-out of NBWs in their respective regions. Training was conducted through a formal two-day training (one in Copenhagen, Denmark in 2018 and one in Lyon, France in 2019). Trained facilitators then must follow one or two NBWs as a support facilitator before being able to lead a workshop. As of 16 July 2020, 10 facilitators are able to lead a NBW, and 22 more can act as support facilitators. A Facilitator's Manual (S2 Appendix) and Facilitator's Checklist-kit (S3 Appendix) were developed and all NBW materials have been standardized to ensure consistent messaging.

An advocacy tool-kit was also produced to raise awareness on this tool, including the NBW Fact-sheet (S3 Appendix), various advocacy videos as well as presentations and posters presented in numerous regional or international conferences and meetings.

## Results

Key lessons learned from the two phases of iterative development of the tool include (i) the need to have a shared understanding of sector-specific assessments such as IHR and PVS and how they contribute to collaborative advantages, (ii) the need to have representatives from different levels (national, sub-national, local) to jointly share the current status of collaboration and discuss how to operationalize shared outputs; (iii) the need for stakeholders to engage as early as possible in scenario-based exercises, so that the conceptualization of joint activities is facilitated and gaps can easily be identified and discussed; (iv) the importance of having the two sectors develop and commit to a joint, realistic and operational roadmap to improve their collaboration; and (v) a well-structured approach and robust facilitation are required for these events.

### Final NBW material and method

The final process of the NBW was split into seven sessions (Table 2) over the course of a three-day in-person workshop and is designed to facilitate engagement with 50-to-90 participants. The objective is to ensure equal representation from both sectors, with participants from national, regional and field levels. Other relevant stakeholders, such as officials from the

**Table 2. Summary of the content and outputs for each session of the NBW.**

| Session | Content | Output |
|---------|---------|--------|
| Session 1 | • Presentations from both sectors<br>• Video on One Health & Tripartite<br>• Video on successful One Health interactions | • Better knowledge of the other sector<br>• Shared understanding of the event's objective |
| Session 2 | • PDWG: Discussion around short scenarios and evaluation of the current collaboration | • Strengths and weaknesses of the collaboration are identified for 15 key technical areas and 4–5 priority diseases |
| Session 3 | • Video and discussion on IHR, SPAR & JEE<br>• Video and discussion on PVS<br>• PDWG: Mapping of the cards identified in session 2 on the large IHR-PVS matrix & discussion | • Better understanding of the two sector-specific frameworks and assessment tools<br>• Priority areas where collaboration needs to be strengthened are identified |
| Session 4 | • TAWG: Extraction of pertinent information from SPAR/JEE, PVS and other relevant assessment reports | • Key gaps and recommendations from sector-specific frameworks are extracted and discussed |
| Session 5 | • TAWG: Brainstorm on joint activities | • A initial, raw joint roadmap is starting to emerge |
| Session 6 | • TAWG: Fine-tuning of activities and detailing of their implementation process<br>• World Café where each group circulates to provide feedback on the other groups' activities<br>• Prioritization exercise | • The joint roadmap is finalized and prioritized |
| Session 7 | • Discussion on the way forward and next steps<br>• Any other working group exercise as per the country's context and needs | • Ownership of the roadmap by the country<br>• Buy-in and leadership on its future implementation<br>• (Optional: anchoring of the roadmap into a higher mandated national plan)<br>• (Optional: other possible collaborative needs are addressed) |

PDWG = Priority disease working group / TAWG = Technical area working group.

**Table 3. Example of Session 2 results from NBW Bhutan.**

| Technical area (cards) | Rabies | Anthrax | H5N1 | Brucellosis | Salmonellosis |
|---|---|---|---|---|---|
| Coordination at high Level | 2 | 2 | 3 | 2 | 2 |
| Coordination at local Level | 2 | 2 | 2 | 2 | 2 |
| Coordination at technical Level | 2 | 2 | 2 | 2 | 2 |
| Legislation / Regulation | 2 | 2 | 3 | 2 | 3 |
| Finance | 1 | 1 | 2 | 2 | 1 |
| Emergency funding | 3 | 2 | 2 | 2 | 2 |
| Communication w/ media | 2 | 2 | 1 | 1 | 2 |
| Communication w/ stakeholders | 2 | 2 | 2 | 3 | 3 |
| Field investigation | 3 | 1 | 2 | 3 | 1 |
| Response | 2 | 3 | 2 | 2 | 2 |
| Risk assessment | 1 | 2 | 2 | 1 | 1 |
| Joint surveillance | 2 | 2 | 2 | 1 | 1 |
| Laboratory | 3 | 3 | 3 | 3 | 2 |
| Education and training | 1 | 1 | 2 | 2 | 1 |
| Human resources | 2 | 3 | 2 | 2 | 1 |
| Logistics | 2 | 2 | 3 | 2 | 1 |

The collaboration for each of the 15 areas was assessed on a 1–3 Likert scale (1/green meaning 'very satisfactory collaboration'; 2/yellow meaning 'some level of collaboration but improvements are needed' and 3/red meaning 'the level of collaboration is really unsatisfactory').

environmental ministry, or observers from collaborating organizations and agencies may also be invited to join, as deemed relevant by the country.

**Session 1** serves as an introduction, with short videos presenting the concept and history of One Health, and with presentations from both sectors to better introduce themselves (their structure, priorities, capacities, etc.) to each other.

In **Session 2**, participants are divided into four or five disease groups. Diseases are chosen in discussion with both Ministries, according to the local context and their priorities. Participants use a fictitious outbreak scenario as a base to discuss how they would realistically manage the situation. In doing so, they must evaluate, using a deck of cards, the level of their collaboration for 15 important technical areas (Table 3) on a three-level Likert scale. This exercise was shown to be very successful in breaking the ice between the different sectors and levels, and in the identification of strengths and weaknesses in the current collaboration.

**Session 3** starts with videos presenting the IHR and related assessment tools (SPAR and JEE) as well as the PVS Pathway (PVS Evaluation and PVS Gap Analysis). Participants are then asked to map the cards that they have selected in the previous session on a 5x3 meter matrix, built with the indicators of the SPAR/JEE and the PVS Pathway (Fig 1). This step allows participants to realize the amount of commonality between the two sectors and their respective frameworks. It also allows for a better visualization of the overall strengths and weaknesses of the collaboration with all priority diseases considered. The collective analysis of the results enables the identification of four or five technical groups for the next exercises to focus efforts on the key technical areas showing the most important gaps. To tackle a maximum of areas, newly-formed groups often address two of the technical cards, such as 'Surveillance' and 'Laboratory' or 'Response' and 'Outbreak Investigation'.

In **Session 4**, each newly formed technical group opens the PVS Evaluation and SPAR or JEE reports and extracts the key findings that are relevant for their area by completing Gap and Recommendation cards.

In **Session 5**, each group compiles all the information collected in sessions 2, 3 and 4 and starts to brainstorm on SMART (specific, measurable, achievable, realistic and time-bound) joint activities that should be conducted to fill the identified gaps and to improve the collaboration between the two sectors in their technical area of focus. The NBW roadmap starts to take shape.

**Session 6** is about structuring and going further into the description of the activities to make them as operational as possible. Groups are given Activity cards that they must fill for each activity. The card asks for a detailed description of the activity, who will be leading its implementation, what will be the exact step-by-step implementation process and what is the desired achievement date. At this stage, exchanges with the facilitating team to help organize, structure and detail the different activities is essential. To facilitate future prioritization, the feasibility and impact of each activity is assessed by participants on a three-level Likert scale. Finally, a world café exercise is organized: the different groups rotate to consider the other groups' boards and are given 15 minutes to provide comments, suggestions or edits. This peer-reviewing process ensures that participants can contribute to all technical areas while also improving the quality of the final road-map. A quick prioritization is then conducted during which each participant must choose the 5 activities considered of highest priority (either through an electronic vote using Google Forms or by posting stickers on the Activity cards directly). At this stage, the roadmap is considered complete (Fig 2).

**Session 7** is the final session and is less standardized than the six previous ones. It aims for several objective: (i) to obtain the buy-in of the roadmap by both sectors, (ii) to ensure that the country takes ownership of the workshop's output, (iii) to discuss on how the roadmap will be implemented and (iv) when possible, to anchor the roadmap in an existing mandated plan.

| Action | Timeline | Difficulty (1-3 scale) | Impact (1-3 scale) | Responsibility | Process |
|---|---|---|---|---|---|
| **JOINT RISK ASSESSMENT & JOINT SURVEILLANCE** | | | | | |
| **Objective 1: Build capacities to strengthen the surveillance system and sharing of information between both sectors** | | | | | |
| **1.1 Establish a joint surveillance working (sub-)group (JSWG) at national (ministerial) and institutional levels** | June 2020 | ++ | +++ | Joint Working Group on Zoonoses (JWGZ), Department for Public Health of MoH (DPH), Department for Animal Health of MoA (DAH) | 1) Establish joint surveillance working groups at national and institutional levels 2) Develop ToR for JSWGs at all levels 3) Develop working plans for JSWGs at all levels 4) Nominate members of JSWGs (six experts including chairman at the national level) |
| **1.2 Develop an electronic surveillance system for the public health sector and integrate it with existing electronic surveillance system for the animal health sector at all health care levels** | April 2021 | +++ | +++ | JSWG, DPH, DAH | 1) The integrated electronic surveillance system should ensure routine sharing of data related to priority zoonoses 2) National JSWG to agree on the type and format of data to be shared between the sectors 3) National JSWG to develop technical specifications including databases, interface, incorporation of GIS, etc. 4) Tender an IT company 5) Develop and test the electronic system 6) Legitimize and implement 7) Train relevant personnel at all levels |
| **Objective 2: Harmonize national surveillance system** | | | | | |
| **2.1 Identify priority zoonotic diseases of joint concern** | 2021 | + | ++ | JSWG, DPH, DAH | 1) Develop concept note 2) Develop/adapt methodology (encountering results of strategic risk assessment (activity 4.3) 3) Conduct a joint workshop on prioritization of zoonotic diseases 4) Prepare workshop report and approve by both sectors |
| **2.2 Revise the operational framework for evidence-based surveillance in both sectors** | April 2020 | + | ++ | JSWG, DPH, DAH, Veterinary Institutes Belgrade, IPHS Batut | 1) To prepare a draft version of an operational framework conduct a meeting with six representatives, three from each sector: a. one representative from each ministry, MoH and MoA, b. two representatives from the epizootiology units from Veterinary Institutes Belgrade and c. two epidemiologists from IPHS Batut 2) Clearly define an operational framework with terms of reference that will be applicable in both sectors |

**Fig 2. Example of an extract from a NBW roadmap (Serbia, November 2019).** The full roadmap contains 11 specific objectives and 27 activities.

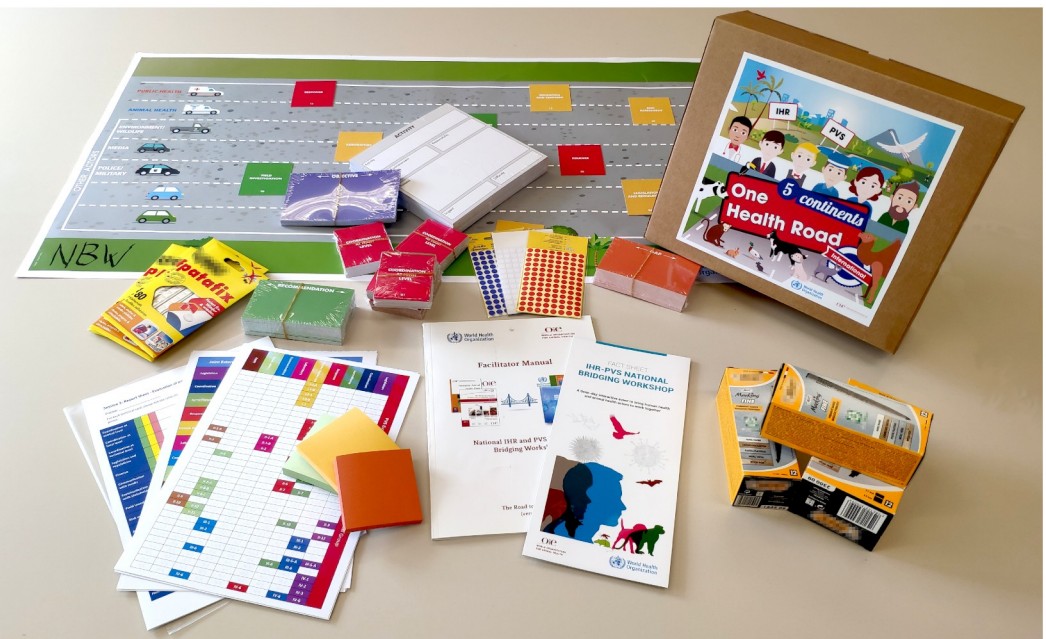

**Fig 3. The NBW material tool-kit comprises posters, technical cards, fact sheets, stationary supplies and a facilitator manual.** The tool-kit is provided by WHO and OIE headquarters. The participant handbooks and assessment (SPAR/JEE, PVS) reports are printed locally.

Facilitators from WHO and OIE withdraw themselves, allowing national/country staff to lead the session and determine next steps for their context. The exact process depends on a country-by-country basis and is planned ahead of the workshop through discussions with a few key national stakeholders. In Bhutan for example, the session was used to inject the activities of the roadmap directly into the national five-year One Health Strategic Plan which was in development. In Pakistan, a federal country, an additional working group exercise was conducted with participants from the same province discussing on how to translate the implementation of the national roadmap at the provincial level. In Indonesia, the two sectors used this opportunity to jointly prepare for the upcoming JEE. In several countries (Jordan, Pakistan, Morocco among others) the session was used to inject the NBW activities into their National Action Plan for Health Security. In Nigeria, another half day was added to extend this final session and use the NBW results to support the creation of a national One Health platform.

The NBW method has been summarized in a video (S1 Video) available at www.bit.ly/NBWMethod.

The NBW material tool-kit (Fig 3) and matrix (Fig 1) exist in English, French, Russian and Spanish versions.

## NBW roll-out

After the six initial pilots conducted between 2013 and 2017 (phase one and two of development), three additional NBWs were conducted in 2017, 11 in 2018, 11 in 2019 and 1 in 2020 (workshops planned in 2020 were cancelled or postponed due to the COVID-19 pandemic), for a total of 32 countries across different regions and continents (Table 4).

The number of participants ranged from 26 (Macedonia) to 85 (Indonesia) with an average of 61, making an aggregate of 1,962 persons who had the opportunity to be engaged in NBWs.

**Table 4. Distribution of NBWs conducted by continent.**

| Africa | Americas | Asia | Europe |
|---|---|---|---|
| Uganda | Costa Rica | Azerbaijan | Albania |
| Tanzania | Belize | Thailand | Armenia |
| Senegal | | Pakistan | Moldova |
| Morocco | | Indonesia | North Macedonia |
| Ethiopia | | Jordan | Serbia |
| Guinea | | Bhutan | Georgia |
| Sierra Leone | | Kyrgyzstan | |
| Chad | | Kazakhstan | |
| Liberia | | Bangladesh | |
| Mauritania | | Myanmar | |
| Niger | | | |
| Benin | | | |
| Nigeria | | | |
| Mali | | | |

A total of 32 countries have conducted a NBW (six pilots between March 2013 and September 2017 followed by 26 workshops between October 2017 and February 2020).

A total of 1,290 participant feedback forms were collected from 28 NBWs. Notably, results show a 97.7% overall satisfaction rate among participants with a 3.5/4 average Likert score. 80.6% of participants declared that the workshop would have a 'Significant' or 'Very High' impact on the improvement of the collaboration between the two sectors in their country. Finally, 99.7% of participants responded that they would recommend this workshop to other countries (Table 5).

The NBW calendar, along with roll-out status by country, and publicly-available NBW reports and roadmaps are available at the following link: https://extranet.who.int/sph/ihr-pvs-bridging-workshop.

**Table 5. Summary of results from 1,290 NBW participant feedback forms.**

| Satisfaction assessment | | |
|---|---|---|
| | Satisfied or Very Satisfied | Average score (Likert scale 1–4) |
| Overall rating | 97.7% | 3.5 |
| Content (Quality, relevance) | 97.4% | 3.5 |
| Structure (Method, material, activities) | 96.4% | 3.5 |
| Facilitators (Communication skills, technical expertise) | 97.7% | 3.6 |
| Organization (Logistics, venue) | 88.5% | 3.4 |
| **Impact assessment** | | |
| | 'Significant' or 'Very High' Impact | Average score (Likert scale 1–4) |
| Impact on participant's technical knowledge | 94% | 3.2 |
| Impact on work of department | 90.1% | 3.3 |
| Impact on AHI collaboration | 80.6% | 3.1 |
| **Recommendation** | | |
| Would you recommend this workshop to other countries? | | 99.7% Yes |

## Discussion

The NBW is a novel tool which bridges internationally accepted framework and tools from the two sectors to allow for improved collaboration while supporting sector-specific needs. It is the first tool that aims to do this and as such, no similar effort or tools was found in the literature for comparison. Our experience in conducting these workshops has shown us that the One Health approach is generally accepted and desired in most countries, but the bottleneck is often in finding out how to adjust the existing systems and habits to concretely operationalize it across both sectors. Because collaboration takes time and energy, it was quickly determined that if One Health efforts could support sector-specific goals and mandates, as shown with IHR (2005) and PVS, they could facilitate the alignment of ongoing activities and a more efficient use of limited resources. In fact, despite the fact that NBW remains a novel tool and that it requires a significant commitment from both sectors (taking 50-to-90 national experts away from their duty for three full days, many of which have to travel long distances to reach the venue), 32 countries, involving a total of 1,962 actors, have already reached out to WHO and OIE to conduct a NBW. This illustrates the strong appetite for One Health and for tools that support its implementation at country level.

In many of those events, officials told us this was the first time that so many stakeholders from the two sectors were meeting to discuss and work specifically on their collaboration. In addition, because the NBWs evolved to include both national and subnational levels, the workshop provided a rare opportunity to amplify the voices at all levels of the human and animal health systems. It was observed that as the discussions unfolded, so did their interest. Participants kept asking for extra time, more sessions and more discussions. For this reason, the overall length of the NBW gradually increased: 1.5 day (Azerbaijan), 2 days (Thailand), 2.5 days (Costa Rica) before reaching its final length of 3 days (Pakistan and all onward workshops). Even with a 3-day process, the most frequent suggestion in the post-workshop surveys was still to increase yet again the duration of the event.

The fact that the 32 workshops had varying levels of success (as judged either from the post-workshop survey or from our own impression) provided essential clues on key success factors to consider: (i) high-level engagement and country ownership, (ii) participant representation, (iii) interactive and participatory approach with robust facilitation and (iv) linkages with IHR and PVS sector-specific goals.

Political will and leadership with sturdy government support and sustainable funding mechanisms are essential for the institutionalization of One Health in countries [22, 34, 35]. The fact that Ministries reach out to WHO and OIE for a NBW and are ready to commit a significant portion of their staff for this three-day event is already a good indication of political commitment. The workshops which we felt were more successful and promising were the ones self-funded by the countries themselves (such as Indonesia or Morocco), perhaps signaling an intention of serious commitment. It is important to clarify the objective and the role that participants are expected to play from the very start of the workshop, and to clearly stress that the NBW is neither a training, nor an external evaluation. Evidence shows that when it comes to operationalizing One Health, there is no one-size fits all approach, and the differences between countries, their health systems, their organizations and their cultures forbid any top-down prescription of measures [14]. The aim is to bring a robust and tested methodology that creates a conducive environment for national staff to identify and discuss their needs themselves (not based on any standards or universal scale of progress) and to derive bespoke solutions, tailored to the country's structure and challenges.

The buy-in and sense of ownership of the resulting road-map is also critical for the improvement of the collaboration at medium and long-terms and a few select focal points

from both sectors, involved very early in the preparation process, are often instrumental for this purpose. Despite the fact the workshop follows a specific methodology, some adjustments to better fit the local context and culture are often made. The national focal points for the NBW organization are also engaged in the design of the simulation scenarios, and often play the role of moderators in the working groups. Whenever possible, they also act as chairperson during the workshop, alternating between the two sectors through the different sessions. The seventh and last session is usually entirely led by the country's national focal points, with OIE and WHO facilitators standing back as discussions are held on the way forward and on the ownership and future implementation of the roadmap. Finally, another important point for the uptake of the roadmap is to make sure, whenever possible, to anchor it into another already mandated plan benefiting for a strong political will and sturdy momentum. For example, in Jordan and Pakistan among others, the activities of the NBW roadmap were injected into the National Action Plan for Health Security, and in Bhutan, Kazakhstan or Nigeria, the joint activities identified during the NBW were anchored into their One Health Strategic Plan.

Because of the very active role that they play throughout the workshop, the selection of participants is a critical factor for success. By experience, the ideal audience size is around 60 participants, with about half from each sector as well as a few representatives of other relevant sectors (wildlife, environment, law enforcement, etc.).

Besides the number, the distribution of participants is also essential. As we know that challenges of One Health operationalization are often found at the local or subnational level [36], it is important that the representatives from each sector originate from the different levels of administration: mainly national, sub-national and local levels. This mixed distribution of sectors and levels is critical, not only for the overall participation, but also for each working group in the different exercises as it allows a diversity of point-of-views throughout the chain of command and throughout the territory. Without this, there is a risk that the identification of gaps and the planned measures in the roadmap remain very superficial and conceptual.

The One Health approach is often visualized with three key actors: human health, animal health and environmental health [15, 18, 20–23, 34, 35]. Several reasons can explain why the latter is not more significantly represented in NBWs: (i) there is no regulatory framework similar to the IHR or the OIE's Terrestrial Animal Health Code upon which to base the workshop; (ii) there is no assessment tool that could be used during the process similar to WHO's SPAR/JEE or to OIE's PVS Evaluation; and (iii) evaluating gaps and identifying ways to improve the collaboration between three separate entities becomes complicated, as was experienced in one workshop where we attempted a NBW with equal number of participants from the three sectors.

In addition to these upstream factors, some downstream efforts are also made to ensure adequate and sustainable follow-up of this initiative in countries. Firstly, the Tripartite—WHO, OIE and FAO—provides implementation guidance [37] and operational tools [38] to support countries in concretizing One Health principles. Secondly, the Tripartite has initiated in 2020 the *NBW Follow-up Program* which includes the recruitment of nationally-hired focal points called *NBW Sherpas*. Their tasks will include, among others, (i) keeping the momentum alive after the NBW by maintaining the liaison between the two sectors; (ii) monitoring, promoting and catalyzing the implementation of the roadmap activities; (iii) providing technical support; and (iv) serving as a relay for other Tripartite One Health tools and activities in countries. The first NBW Sherpas are due to be hired in January 2021.

## Conclusion

In an increasingly complex and globalized world, with competing priorities, the One Health approach is becoming more and more relevant. As national governments seek to strengthen

their capacity for zoonotic disease prevention, detection and response, they need tools to both diagnose needs and existing gaps, as well as develop action plans to support collaboration across sectors. The NBW process, as developed through a series of pilots, supports countries to link their inter-sectoral goals to existing international standards and assessments such as the OIE PVS Pathway and the WHO SPAR/JEE. The ability to collaborate while supporting sector-specific needs provides added incentives for ongoing and sustainable collaborations at the human-animal interface.

## Supporting information

**S1 Appendix. NBW participant handbook.**
(PDF)

**S2 Appendix. NBW facilitator manual.**
(PDF)

**S3 Appendix. NBW facilitator's checklist-kit.**
(PDF)

**S4 Appendix. NBW fact sheet.**
(PDF)

**S1 Video. NBW method overview.**
(WMV)

## Acknowledgments

We would like to sincerely thank all national stakeholders from the 32 countries who have contributed to the planification, organization and running of their NBW, as well as staff in the WHO regional and country offices who have been instrumental in the organization of those workshops.

## Author Contributions

**Conceptualization:** Guillaume Belot, François Caya, Susan Corning, Rajesh Sreedharan, Jun Xing, Stephane de la Rocque.

**Data curation:** Guillaume Belot, Tieble Traore, Artem Skrypnyk.

**Funding acquisition:** François Caya, Tieble Traore, Brice Lafia, Nicolas Isla, Gyanendra Gongal, Stella Chungong, Stephane de la Rocque.

**Methodology:** Guillaume Belot, François Caya, Susan Corning, Rajesh Sreedharan, Gyanendra Gongal, Enrique Perez-Gutierrez, Stephane de la Rocque.

**Project administration:** Guillaume Belot, François Caya, Tieble Traore, Brice Lafia, Artem Skrypnyk, Djhane Montabord, Maud Carron, Nicolas Isla, Tanja Schmidt, Gyanendra Gongal, Dalia Samhouri, Ana Riviere-Cinnamond, Stephane de la Rocque.

**Resources:** Guillaume Belot, Stephane de la Rocque.

**Supervision:** Stella Chungong, Stephane de la Rocque.

**Validation:** Stephane de la Rocque.

**Writing – original draft:** Guillaume Belot, Kaylee Myhre Errecaborde, Stephane de la Rocque.

**Writing – review & editing:** Guillaume Belot, François Caya, Tieble Traore, Artem Skrypnyk, Djhane Montabord, Maud Carron, Susan Corning, Rajesh Sreedharan, Nicolas Isla, Tanja Schmidt, Gyanendra Gongal, Dalia Samhouri, Jun Xing.

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
