## [Decision Letter · Decision Letter 0]

17 Feb 2021

PONE-D-20-40093

IHR-PVS National Bridging Workshops, a tool to operationalize the collaboration between human and animal health while advancing sector-specific goals in countries

PLOS ONE

Dear Dr. BELOT,

Thank you for submitting your manuscript to PLOS ONE. After careful consideration, we feel that it has merit but does not fully meet PLOS ONE’s publication criteria as it currently stands. Therefore, we invite you to submit a revised version of the manuscript that addresses the points raised during the review process.

Please pay attention to the introduction section, some revisions are suggested by the reviewers.

Reviewers have also suggested to revise results and discussion sections comprehensively.

Please modify Figures for more clarity and better understanding.

We look forward to receiving your revised manuscript.

Kind regards,

Ghaffar Ali, PhD

Academic Editor

PLOS ONE

Journal Requirements:

"This work has been supported by many funders including the United States Defense Threat

 Reduction Agency (US-DTRA), the Global Partnership Program (GPP), the EU Commission's

Directorate-General for International Cooperation and 475 Development (DG DEVCO), the Russian

Federation and the World Bank, among others."

"The authors received no specific funding for this work"

3. Please remove your figures from within your manuscript file, leaving only the individual TIFF/EPS image files, uploaded separately.  These will be automatically included in the reviewers’ PDF.

4. We note that Figure 4 in your submission contain map images which may be copyrighted. All PLOS content is published under the Creative Commons Attribution License (CC BY 4.0), which means that the manuscript, images, and Supporting Information files will be freely available online, and any third party is permitted to access, download, copy, distribute, and use these materials in any way, even commercially, with proper attribution. For these reasons, we cannot publish previously copyrighted maps or satellite images created using proprietary data, such as Google software (Google Maps, Street View, and Earth). For more information, see our copyright guidelines: http://journals.plos.org/plosone/s/licenses-and-copyright.

4.1.    You may seek permission from the original copyright holder of Figure 4 to publish the content specifically under the CC BY 4.0 license. 

4.2.    If you are unable to obtain permission from the original copyright holder to publish these figures under the CC BY 4.0 license or if the copyright holder’s requirements are incompatible with the CC BY 4.0 license, please either i) remove the figure or ii) supply a replacement figure that complies with the CC BY 4.0 license. Please check copyright information on all replacement figures and update the figure caption with source information. If applicable, please specify in the figure caption text when a figure is similar but not identical to the original image and is therefore for illustrative purposes only.

Reviewers' comments:

Reviewer's Responses to Questions

**Comments to the Author**

1. Is the manuscript technically sound, and do the data support the conclusions?

Reviewer #1: Yes

Reviewer #2: Yes

2. Has the statistical analysis been performed appropriately and rigorously? 

Reviewer #1: I Don't Know

Reviewer #2: Yes

3. Have the authors made all data underlying the findings in their manuscript fully available?

Reviewer #1: Yes

Reviewer #2: Yes

4. Is the manuscript presented in an intelligible fashion and written in standard English?

Reviewer #1: No

Reviewer #2: No

5. Review Comments to the Author

Reviewer #1: The enjoyed reading this article unveiling the role of National Bridging Workshops in operationalizing the collaboration between human and animal health sectors. The authors have highlighted the importance of integration of sector-specific and collaborative goals. The article can be considered for publication in the PLOS ONE. I have the following minor comments.

1. Introduction: Authors may consider revising the line 87-92.

2. Results: Authors should provide a summary of the key finding and significant features of the tool before getting into the detailed description. It is difficult to get the salient outcomes of the study in its current form.

3. Fig 4. Please indicate what different colours are referring to. I would suggest adding country names as well which make it easier to understand for the readers with different parts of the world.

4. The discussion could have been improved by incorporating major outcomes and its relevant literature from similar efforts/tools developed in the past.

5. Currently, methods, results and discussion – all the sessions are amalgamated and twisting around.

6. Conclusion: No comment.

Reviewer #2: The One Health approach is often visualized with three key actors: human health, animal health and

environmental health. The authors explains in the emanuscript an endeavour for one health program across nations through a series of six phased pilots, the IHR-PVS National Bridging Workshop (NBW) method was developed and refined. The NBW process gathers human and animal health stakeholders and follows seven sessions, scheduled across three days. The outputs from each session build towards the next one, following a structured process that goes from gap identification to joint planning of corrective measures. These efforts helped identify the strength and gaps in country's preparedness. The NBW process allows human and animal health sector representatives to jointly identify actions that support collaboration while advancing evaluation goals identified through the IHR-MEF and the OIE PVS Pathway. These paper elaborates how an international effort was used counter One health issues and program to address the gaps was launched. The authors should carry more such programs in regard to One Health.

6. PLOS authors have the option to publish the peer review history of their article (what does this mean?). If published, this will include your full peer review and any attached files.

Reviewer #1: No

Reviewer #2: No

---

## [Author Response · Author response to Decision Letter 0]

7 Apr 2021

15 March 2021: Upon request from the journal, authors updated the ethic statement to the following:

"No research was conducted on human subjects or other animal subjects for the purpose of this article therefore no ethics approval was required. Participants to the workshops were invited and came in full consent. Their consent was not documented in any written way. Participants were informed as to the nature of their participation (fact-sheet, concept note, agenda) prior to coming to the workshop. In the opening session of every workshop, the first presentation gave an overview of the method and process of the workshop and stated that results would be compiled in a report and posted on the WHO website and may later be used for further research and publication. The information obtained was recorded by the investigator in such a manner that the identity of the Human Subjects cannot be readily ascertained, directly or indirectly through identifiers linked to the subjects."

Journal Requirements:

Response from authors: after revision of the style requirements, the following corrections were brought to the manuscript:

• Removal of Figures from the manuscript

• Relocation of the legends after the tables for Table 2 and Table 3

• Relocation of the ‘Supporting Information’ section after the ‘References’ section

The manuscript now meets PLOS ONE’s style requirements as detailed in the templates provided.

"This work has been supported by many funders including the United States Defense Threat

 Reduction Agency (US-DTRA), the Global Partnership Program (GPP), the EU Commission's

Directorate-General for International Cooperation and 475 Development (DG DEVCO), the Russian

Federation and the World Bank, among others."

"The authors received no specific funding for this work"

Response from authors:

• The funding-related paragraph was removed from the Acknowledgments section.

• Our amended statement was added in the cover letter: “The organization of IHR-PVS National Bridging Workshops in countries was supported by many funders including the United States Defense Threat Reduction Agency (US-DTRA), the Global Partnership Program (GPP), the EU Commission's Directorate-General for International Cooperation and Development (DG DEVCO), the Russian Federation and the World Bank, among others.”

3. Please remove your figures from within your manuscript file, leaving only the individual TIFF/EPS image files, uploaded separately. These will be automatically included in the reviewers’ PDF.

Response from authors: All figures have been removed from the manuscript accordingly.

 4. We note that Figure 4 in your submission contain map images which may be copyrighted. All PLOS content is published under the Creative Commons Attribution License (CC BY 4.0), which means that the manuscript, images, and Supporting Information files will be freely available online, and any third party is permitted to access, download, copy, distribute, and use these materials in any way, even commercially, with proper attribution. For these reasons, we cannot publish previously copyrighted maps or satellite images created using proprietary data, such as Google software (Google Maps, Street View, and Earth). For more information, see our copyright guidelines: http://journals.plos.org/plosone/s/licenses-and-copyright.

We require you to either (1) present written permission from the copyright holder to publish these figures specifically under the CC BY 4.0 license, or (2) remove the figures from your submission.

Response from authors: The map (Figure 4) was replaced by a table (Table 4) for enhanced clarity and to avoid any potential copyright issue.

Comments from Reviewers:

Reviewer #1

Reviewer #1: The enjoyed reading this article unveiling the role of National Bridging Workshops in operationalizing the collaboration between human and animal health sectors. The authors have highlighted the importance of integration of sector-specific and collaborative goals. The article can be considered for publication in the PLOS ONE. I have the following minor comments.

1. Introduction: Authors may consider revising the line 87-92.

2. Results: Authors should provide a summary of the key finding and significant features of the tool before getting into the detailed description. It is difficult to get the salient outcomes of the study in its current form.

3. Fig 4. Please indicate what different colours are referring to. I would suggest adding country names as well which make it easier to understand for the readers with different parts of the world.

4. The discussion could have been improved by incorporating major outcomes and its relevant literature from similar efforts/tools developed in the past.

5. Currently, methods, results and discussion – all the sessions are amalgamated and twisting around.

6. Conclusion: No comment.

Response from authors:

1. As suggested, lines 87-92 in the introduction have been fully rewritten and expanded upon for enhanced clarity (lines 93-98 of the track changes version).

2. The paragraph summarizing the 5 key lessons learned from the development of the tool was added in the Results section (lines 264-272) right before the detailed description: “Key lessons learned from the two phases of iterative development of the tool include (i) the need to have a shared understanding of sector-specific assessments such as IHR and PVS and how they contribute to collaborative advantages, (ii) the need to have representatives from different levels (national, sub-national, local) to jointly share the current status of collaboration and discuss how to operationalize shared outputs; (iii) the need for stakeholders to engage as early as possible in scenario-based exercises, so that the conceptualization of joint activities is facilitated and gaps can easily be identified and discussed; (iv) the importance of having the two sectors develop and commit to a joint, realistic and operational roadmap to improve their collaboration; and (v) a well-structured approach and robust facilitation are required for these events.”

3. The map (Figure 4) was replaced by a table (Table 4) displaying all the country names for clearer understanding.

4. We agree with the reviewer statements and have clarified and highlighted the following key outcomes in lines 389-391 of the version with track changes: The NBW is a novel tool which bridges internationally accepted framework (IHR and PVS) and tools from the two sectors to allow for improved collaboration while supporting sector-specific needs. It is the first tool that aims to do this and as such, no similar effort or tools was found in the literature for comparison.

5. Method: The authors have made every attempt to clarify the structure and flow of the paper to walk to the reader through a linear progression of methods, results and discussion. The method section now describes the iterative development of the process of NBWs, which of course was an iterative methodology of improvement. Once the full method and material for the NBWs were finalized, the process was divided into 7 distinct sessions. We edited line 254 to better clarify this: “The final process of the NBW was split into seven sessions (Table 2) over the course of a three-day in-person workshop”.

Result: We made some formatting changes to better separate the different sessions and make them more visible. Each session has its own paragraph to avoid any blending and to make sure each is distinctive. In addition, we moved the introduction of key findings from the methods to results as was suggested under point 2, see lines 285-293.

Discussion: The discussion points address the NBW event as a whole and are not session-specific. We discuss the fact that it is the first event of this type and then discuss what from our experience were the key success factors: (i) high-level engagement and country ownership, (ii) participant representation, (iii) interactive and participatory approach with robust facilitation and (iv) linkages with IHR and PVS sector-specific goals. These four points are addressed in further details in the manuscript but they are generic to the whole workshop and not specific to any session. To avoid any confusion, we removed any reference to a session number, apart from the paragraph detailing the role of country officials in the last session of the workshop as it is an important example to discuss point (i). As recommended in reviewer point 4, we added additional language to highlight the key outcome in lines 438-440. 

Reviewer #2

Reviewer #2: The One Health approach is often visualized with three key actors: human health, animal health and environmental health. The authors explains in the emanuscript an endeavour for one health program across nations through a series of six phased pilots, the IHR-PVS National Bridging Workshop (NBW) method was developed and refined. The NBW process gathers human and animal health stakeholders and follows seven sessions, scheduled across three days. The outputs from each session build towards the next one, following a structured process that goes from gap identification to joint planning of corrective measures. These efforts helped identify the strength and gaps in country's preparedness. The NBW process allows human and animal health sector representatives to jointly identify actions that support collaboration while advancing evaluation goals identified through the IHR-MEF and the OIE PVS Pathway. These paper elaborates how an international effort was used counter One health issues and program to address the gaps was launched. The authors should carry more such programs in regard to One Health.

Response from authors: No specific revision was requested by Reviewer #2.

---

## [Decision Letter · Decision Letter 1]

4 May 2021

IHR-PVS National Bridging Workshops, a tool to operationalize the collaboration between human and animal health while advancing sector-specific goals in countries

PONE-D-20-40093R1

Dear Dr. BELOT,

We’re pleased to inform you that your manuscript has been judged scientifically suitable for publication and will be formally accepted for publication once it meets all outstanding technical requirements.

Kind regards,

Ghaffar Ali, PhD

Academic Editor

PLOS ONE

Additional Editor Comments (optional):

Reviewers' comments:

Reviewer's Responses to Questions

**Comments to the Author**

1. If the authors have adequately addressed your comments raised in a previous round of review and you feel that this manuscript is now acceptable for publication, you may indicate that here to bypass the “Comments to the Author” section, enter your conflict of interest statement in the “Confidential to Editor” section, and submit your "Accept" recommendation.

Reviewer #1: All comments have been addressed

Reviewer #2: All comments have been addressed

2. Is the manuscript technically sound, and do the data support the conclusions?

Reviewer #1: Yes

Reviewer #2: Yes

3. Has the statistical analysis been performed appropriately and rigorously? 

Reviewer #1: Yes

Reviewer #2: N/A

4. Have the authors made all data underlying the findings in their manuscript fully available?

Reviewer #1: Yes

Reviewer #2: Yes

5. Is the manuscript presented in an intelligible fashion and written in standard English?

Reviewer #1: Yes

Reviewer #2: Yes

6. Review Comments to the Author

Reviewer #1: Authors have addressed all the comments and I am now happy to accept the manuscript for publication.

Reviewer #2: (No Response)

7. PLOS authors have the option to publish the peer review history of their article (what does this mean?). If published, this will include your full peer review and any attached files.

Reviewer #1: No

Reviewer #2: No

---

## [Editor Report · Acceptance letter]

17 May 2021

PONE-D-20-40093R1 

IHR-PVS National Bridging Workshops, a tool to operationalize the collaboration between human and animal health while advancing sector-specific goals in countries 

Dear Dr. Belot:

I'm pleased to inform you that your manuscript has been deemed suitable for publication in PLOS ONE. Congratulations! Your manuscript is now with our production department. 

Kind regards, 

on behalf of

Dr. Ghaffar Ali 

Academic Editor

PLOS ONE